# An Overview of Biofilm-Associated Infections and the Role of Phytochemicals and Nanomaterials in Their Control and Prevention

**DOI:** 10.3390/pharmaceutics16020162

**Published:** 2024-01-24

**Authors:** Tsvetozara Damyanova, Petya D. Dimitrova, Dayana Borisova, Tanya Topouzova-Hristova, Emi Haladjova, Tsvetelina Paunova-Krasteva

**Affiliations:** 1Stephan Angeloff Institute of Microbiology, Bulgarian Academy of Sciences, Akad. G. Bonchev St. bl. 26, 1113 Sofia, Bulgaria; tsvetozaradamianova@gmail.com (T.D.); pdimitrova998@gmail.com (P.D.D.); daqanara@abv.bg (D.B.); 2Faculty of Biology, Sofia University “St. K. Ohridski”, 8 D. Tsankov Blvd., 1164 Sofia, Bulgaria; 3Institute of Polymers, Bulgarian Academy of Sciences, Akad. G. Bonchev St. bl. 103-A, 1113 Sofia, Bulgaria; ehaladjova@polymer.bas.bg

**Keywords:** biofilms, biofilm-associated infections, nanomaterials, phytochemicals

## Abstract

Biofilm formation is considered one of the primary virulence mechanisms in Gram-positive and Gram-negative pathogenic species, particularly those responsible for chronic infections and promoting bacterial survival within the host. In recent years, there has been a growing interest in discovering new compounds capable of inhibiting biofilm formation. This is considered a promising antivirulence strategy that could potentially overcome antibiotic resistance issues. Effective antibiofilm agents should possess distinctive properties. They should be structurally unique, enable easy entry into cells, influence quorum sensing signaling, and synergize with other antibacterial agents. Many of these properties are found in both natural systems that are isolated from plants and in synthetic systems like nanoparticles and nanocomposites. In this review, we discuss the clinical nature of biofilm-associated infections and some of the mechanisms associated with their antibiotic tolerance. We focus on the advantages and efficacy of various natural and synthetic compounds as a new therapeutic approach to control bacterial biofilms and address multidrug resistance in bacteria.

## 1. Introduction

Biofilm-associated infections represent a major challenge for clinicians due to their high prevalence rate of over 80%. In order to enhance the quality of life of the aging population, medical implant procedures are increasingly required. However, their use is associated with the development of biofilm or nosocomial infections, most of which are chronic and unfortunately incurable. Infections associated with the development of bacterial biofilms can occur on both implantable and non-implantable medical devices, such as cardiac pacemakers, heart valves, artificial joints, venous and urinary catheters, endotracheal tubes, breast implants, contact lenses, intrauterine devices, biliary stents, orthodontic prostheses, and more [1,2,3,4]. In addition, there are infections connected with conditions unrelated to implants, including cystic fibrosis, chronic obstructive pulmonary diseases, otitis, infectious wounds, diabetes, sinusitis, osteomyelitis, endocarditis, chronic prostatitis, and others [5,6,7,8,9]. Low bacterial sensitivity and resistance to conventional therapeutics, including antibiotics, and bacteria’s ability to invade and spread contribute to high mortality rates among patients, becoming a significant public health concern. Bacteria in biofilms, compared to planktonic cells, are significantly more resistant to antimicrobial agents, which is also one of the reasons for the appearance of chronic and persistent infections. Additionally, extracellular polymeric substances (EPSs) produced by biofilms promote the resistance of pathogenic bacteria to the host’s adaptive and innate immune systems. Microbiological diagnosis and appropriate treatment are crucial for addressing biofilm infections, and they present a significant scientific challenge. In this review, we examined and summarized data related to the colonization, distribution, etiology, and species diversity of biofilm-related infections, along with innovative approaches to their treatment. We emphasized the potential of nanomaterial structures and natural compounds of plant origin as promising alternatives for prevention and treatment.

## 2. Biofilm-Associated Infections in Human Tissues

### 2.1. Bacterial Biofilms in Chronic Wounds

Numerous pieces of evidence suggest that the presence of chronic wounds and their challenging healing can be attributed to the colonization of various microorganisms and the subsequent formation of biofilms [7,10,11,12,13]. Frequently, untreated wounds lead to several complications such as infection, chronicity, morbidity, and amputation. Wounds unable to heal after one month are diagnosed as chronic, and their spread leads to increased morbidity and even mortality [5,14,15]. Furthermore, studies monitoring chronic wounds have revealed that about 60% of them contain biofilms. Similar findings are reported in patients with obesity, diabetes, pressure ulcers, venous leg ulcers, and others [16,17,18,19,20]. Diabetic patients are at increased risk of developing chronic wounds, especially in the area of the feet (so-called diabetic foot ulcers) [7]. More than 50% of diabetic wounds have been found to be infected [21]. Biofilms isolated from wounds are usually multispecies, composed of more than two microorganisms. The most commonly found bacteria are from the genera *Staphylococcus*, *Enterococcus*, *Corynebacteria*, and *Pseudomonas* [22]. The complex behavior of bacteria in biofilms in chronic wounds is primarily attributed to several factors, including variations in bacterial nutritional requirements, the synthesis of EPSs, delayed collagen production, prolonged inflammatory responses, and the production of various types of exotoxins that result in tissue damage and the suppression of cell metabolism. At the same time, in patients with diabetes, fluctuations in blood sugar levels, obstruction of the blood flow carrying the necessary nutrients, oxidative stress, hypoxia, the presence of biofilms, and the existence of different microenvironments significantly hinder the inflammatory response and the immune system’s effective reaction. Using peptide nucleic acid-based fluorescence in situ hybridization (PNA-FISH) combined with confocal laser scanning microscopy (CLSM), samples from nine chronic venous leg ulcers were examined. The obtained results showed a spatial colonization in the wound, in which *S. aureus* was distributed in the superficial layers, while *P. aeuruginosa* was found in the deeper layers [23,24]. It was suggested that the survival of formed biofilms is influenced by their location, as the accessibility of locally administered compounds to the deeper layers of the biofilms is limited. Similar observations were made when applying ionized nanocrystalline silver dressings in 29 patients with various chronic wounds. In some patients, a reduction in the number of microorganisms found in the superficial dermal layers was observed [25]. The thickness of the EPS is also of critical importance, as it can modulate the pathogenesis of the biofilm. Its components affect the polymorphonuclear leukocyte (PMN) response of the immune system. At the same time, the microenvironment of chronic wounds compared to healthy skin is also of decisive importance for the healing process. Factors like pH and oxygen levels are important to this process as well as to the development of biofilm infection. As chronification increases, the pH of the wound becomes alkaline, while with progression to healing, acidity increases, which also reduces the ability of bacteria to form biofilms [26].

### 2.2. Bacterial Biofilms in Burns

Burn wounds also offer a convenient entry point for biofilm-forming pathogens. The loss of the natural skin layer and the denaturation and coagulation of proteins in the burn area promote microbial colonization and subsequent biofilm infections [27]. Such skin burns trigger the synthesis of amplified cytokines and prostaglandins, the release of vasoactive prostanoids and leukotrienes, and the increased production of lymphokines by macrophages. Additionally, the poor blood supply, anaerobic and acidic environment, and the exudate formation in such wounds significantly facilitate the growth of both planktonic and biofilm-forming pathogenic microorganisms [28]. The list of pathogens associated with wound infections caused by burns is extensive, including Gram-positive and Gram-negative aerobic and anaerobic microorganisms, such as those from the genera *Pseudomonas*, *Serratia*, and *Staphylococcus*, viruses, and fungi like *Candida* [9]. Generally, colonization is also due to infection of the wound with the patient’s microflora during transfer from contaminated surfaces as well as various invasive or non-invasive medical procedures, such as urinary catheter placement, hydrotherapy treatment, use of ventilation, etc. It has been observed that 39% of burn patients who undergo skin grafting develop a wound infection and require regrafting [29]. The wound repair process involves a series of interdependent and tightly regulated mechanisms. The initial phase is the inflammatory phase, during which invading “foreign” cells in the wound are eliminated. This is followed by the proliferation or epithelialization phase, in which fibroblasts are activated to increase the production of collagen and other crucial substances. The final phase involves remodeling, associated with the production of collagen type III, leading to the formation of new tissue [28]. If any of these events in the cascade are disrupted, healing is impeded, proinflammatory cytokine production continues, and the damage process resumes. The final result is the death of neutrophils and other cells and their inability to penetrate the biofilm matrix [30]. It is worth mentioning the increased antibiotic resistance of bacterial biofilms, which has been demonstrated to lead to the restoration of antibiotic resistance following an intervention, resulting in reduced treatment efficacy and challenging wound healing [31].

### 2.3. Cystic Fibrosis (CF)

Cystic fibrosis (CF) is a hereditary disease that affects approximately 160,000 children and adults in Europe, with about 2000 newborns affected by the condition. Undiagnosed patients die at an early age. CF affects multiple organs and systems in patients’ bodies, resulting in symptoms such as the accumulation of mucus in the bronchi, increased blood sugar levels, liver changes, and sterility in men. The disease is caused by a mutation in the CFTR (cystic fibrosis transmembrane conductance regulator) gene, which encodes a protein that functions as an ion channel and regulates the transport of chloride and other ions across cell membranes. The standard method for diagnosing CF is the sweat sodium chloride test [32,33]. Mutations in this gene lead to imbalances in ion transport across epithelial cells, particularly in the respiratory and digestive systems. In the respiratory tract, this leads to difficulty breathing and blockage of the lung alveoli, which increases the risk of infections [34]. The presence of mucus provides an ideal environment for the development of biofilm infections. In CF patients, factors such as reduced environmental pH, mucus accumulation on airway surfaces, and the reduced secretion of chloride interfere with innate immunity in the elimination of bacteria [35]. Among the best-studied pathogens related to CF infections are *P. aeruginosa*, *S. aureus*, *Haemophilus influenzae*, and *Burkholderia cenocepacia*. The lung microbiome in CF patients may also contain other pathogenic species, such as *Achromobacter xylosoxidans*, *Streptococcus milleri*, *Ralstonia* spp., *Pandorea* spp., *Stenotrophomonas maltophilia*, and *Mycobacterium spp.* Additionally, oral bacteria can become part of the bacterial community of the respiratory tract in CF patients. Some of the most common bacterial species are *Rothia mucilaginosa*, *Gemella haemolysans, Prevotella* spp., *Veillonella* spp., and *Fusobacterium* spp. [36]. Airway obstruction makes patients susceptible to colonization by opportunistic pathogens, initially with *S. aureus* and later with *P. aeruginosa*, initiating a harmful cycle of infection, inflammation, and tissue remodeling. If this process is left untreated, this cycle results in a continuous decline in lung function and, ultimately, premature patient death due to respiratory failure [35]. The pathogenesis of *P. aeruginosa* is associated with virulence determinants, such as exotoxins S, T, Y, and U, secreted by the type III secretion system which injects them into eukaryotic cells [34]. Persistent infections and inflammation lead to chronic lung damage and respiratory failure, which is the main cause of mortality in patients with CF [8]. Currently, there is data on the phenotypic diversification and adaptation of *P. aeruginosa* during chronic persistence in the lungs of CF patients, which can be observed when comparing isolates or strains collected at different stages of the disease from the same patient [37,38,39].

### 2.4. Biofilm Infections Associated with the Oral Cavity

Diseases of the oral cavity, such as caries, periodontitis, gingivitis, etc., include a wide range of bacterial genera, such as *Streptococcus*, *Porphyromonas*, *Fusobacterium*, *Veillonella*, *Granulicatella*, *Corynebacterium*, *Rothia*, *Actinomyces*, *Prevotella intermedia*, *Capnocytophaga*, *Neisseria*, and *Haemophilus*, of which the dental biofilm is composed [40]. These bacteria inhabit specific anatomical sites that are inaccessible to macrophages or immune cells. They secrete releasing enzymes, exotoxins, metabolic end products, and virulence factors, thus mediating the development of oral diseases [41,42]. Under normal conditions, the resident oral microflora is balanced and tailored to benefit the host. As a results of the disruption of this homeostasis by several factors, such as poor health, consumption of sugar-rich foods, poor oral hygiene, aging processes, genetic factors, and immunity and social status of the host, more virulent species dominate and cause dental diseases. Dental caries is estimated to affect 60–90% of pupils and 100% of adults, while periodontal diseases impact about 10% of the population. Diseases of the oral cavity are the fourth most expensive disease to treat worldwide [43]. Dental caries occurs on teeth above the gum line (supragingival), and periodontal disease occurs below (subgingival), attacking tooth-supporting tissues. The cause of dental caries is cariogenic plaque. Cariogenic plaque occurs when normally low populations of acidogenic and acidic bacterial species that have been in balance with the oral environment are increased, as well as at high carbohydrate concentrations in the oral cavity [44]. The reason for this is that carbohydrate metabolism leads to plaque acidification (pH < 5), resulting in acid-induced demineralization of enamel and dentin, ultimately leading to cavitation. Cavitation is a late event in the pathogenesis of decay. Periodontal disease involves a cellular inflammatory response in the gingiva and surrounding connective tissue due to bacterial accumulation on the teeth, leading to the loss of collagen attachment between the tooth and bone and subsequent bone loss [18,45]. The best-studied periodontal bacteria are *Actinomyces actinomycetemcomitans*, *Porphyromonas gingivalis*, and *Bacteroides forsythus*, due to their ability to synthesize various virulence factors, including invasins and proteases. Other factors contributing to the appearance of periodontitis are stress, diet, obesity, cardiovascular diseases, osteoporosis, diabetes and rheumatoid arthritis, poor oral hygiene, etc.

## 3. Biofilm-Associated Infections and Microbial Colonization of Medical Devices

The application of medical devices has been shown to improve significantly a patient’s quality of life. Unfortunately, this process is also associated with the appearance of a number of infections, usually caused by biofilms. Biofilms can be formed by one or multiple types of bacteria, and this process also depends on the duration of implant placement. The most common pathogens capable of biofilm formation are the Gram-negative *Klebsiella pneumoniae* (30%), *Burkholderia cepacia* complex (30%), and *P. aeruginosa* (15%), and the Gram-positive *Enterococcus faecalis* (54.5%), *S. aureus* (27.3%), and *Staphylococcus epidermidis* (18.2%) [6].

### 3.1. Biofilm Infections Associated with the Urinary Tract

Some of the most common biofilm infections are those caused by uropathogenic microorganisms that adhere to urinary catheters, leading to urinary tract infections (UTIs) [4]. The treatment of these diseases is difficult due to biofilm formation, cell adhesion, increased antibiotic resistance in the biofilm, and the presence of multiresistant strains. Eighty percent of reported infections are caused by the Gram-negative *E. coli* (UPEC). Other commonly found pathogens are *P. aeruginosa*, *Proteus mirabilis*, *E. faecalis*, *K. pneumoniae*, *Staphylococcus saprophyticus*, and *S. epidermidis* [46,47,48]. A significant part of uropathogens originate from the intestinal flora, but they can migrate to the bladder after colonizing the urethra, leading to bacterial prostatitis and pyelonephritis [49,50]. These infections are often associated with a prolonged use of a catheter in the patient. Catheters, typically made of silicon or latex, are frequently used during surgery to evacuate urine. Cross-sectional studies of silicone catheters used for 8 weeks revealed crystalline biofilm blockage in the central lumen [1]. The adhesion process of UPEC to glycoprotein receptors of urothelial cells is facilitated by the presence of type I pili and the protein FimH, known as virulence determinants responsible for bacterial pathogenicity and biofilm formation [51,52]. After the attachment of UPEC cells, the epithelium internalizes them through its own endocytic mechanisms. Bacterial invasion triggers apoptotic cascades and inflammation in human epithelial cells. The influx of neutrophils into the bladder is enhanced through the CD14-TLR4 pathway, which recognizes bacterial lipopolysaccharides [53,54]. UPEC strains can evade the immune response by stimulating proinflammatory activity or by masking an immunogenic bacterial composition. It is believed that women are more affected by this type of infection than men due to the proximity of the rectum, vagina, and urethra, but the process may also be genetically determined. Acute urinary tract infections, often caused by bacterial biofilm formation, can lead to recurrent infections [55]. In clinical nephrology, biofilms influence the occurrence of kidney stones and dialysis systems [56].

### 3.2. Biofilm Infections Associated with Contact Lenses

The development of contact lens biofilm infections is a common process associated with the development of microbial keratitis [57,58]. Bacterial species such as *Staphylococcus aureus*, *Pseudomonas aeruginosa*, *S. epidermidis*, and certain fungal species like *Serratia marcescens*, *Candida albicans*, *Aspergillus*, and *Fusarium* have been isolated in association with these infections [59,60]. Contact lenses facilitate the transfer of microorganisms to the eye surface, where, under normal conditions, their development is limited. The extent of biofilm adhesion to the lens depends on factors such as hydration status, lens characteristics, electrolyte content, and bacterial species [61]. The close interaction between the lens and the corneal epithelium induces local changes, including hypoxia and hypercapnia, which affect the ability of the epithelium to respond to injury. Tear fluid exchange can be compromised between the anterior and posterior sides of the lens, altering the composition of the tear fluid on the ocular surface and limiting its antimicrobial properties [62]. Early symptoms of biofilm-associated keratitis include redness, pain, sensitivity to light, and vision problems. Eye examination may reveal eyelid edema, conjunctival changes, corneal opacities, and varying degrees of anterior chamber inflammation. The condition can lead to corneal scarring or, in extreme cases, corneal perforation and loss of the eye [63]. Isolates from keratitis caused by contact lenses were studied and it was observed that the microorganisms among them were not common commensals. This may have been a result of preformed biofilms on the lenses or ones developed during storage. This suggests that poor hygiene is also among the risk factors associated with the appearance of keratitis [64].

## 4. Therapeutic Strategies to Combat Bacterial Biofilms

Biofilm control is a complex process and often requires combinations of different strategies and agents to achieve the desired effect. Knowledge of the mechanisms of biofilm formation, as well as the development of innovative approaches and the application of various new agents, are essential for the successful control of biofilm infections. Key approaches involve prevention and treatment. Preventive strategies focus on interrupting biofilm formation processes, while treatment strategies target already formed mature biofilms.

When the goal is prevention, efforts should be directed towards creating conditions that hinder or prevent biofilm formation in its initial stages. This includes actions such as blocking and preventing the adhesion of bacteria to the substrate [65]; blocking the synthesis stages of the biofilm matrix; inhibiting the quorum-sensing signaling cascade; and disrupting of communication between bacteria [66,67]. Treatment as a strategy aims to prevent the biofilm from growing and occupying new niches, which can lead to infections. Consequently, this strategy aims to control and eliminate the formed biofilm [68,69]. Conventional agents like antibiotics are commonly used, but escalating antibiotic resistance has led to a quest for substances with antibacterial or antibiofilm effects [70,71,72,73]. Unfortunately, the use of antibiotics has become a risk factor due to growing antibiotic resistance, underscoring the urgent need to discover more promising approaches to combating multidrug resistance.

## 5. Nanomaterials against Biofilm-Related Infections

The development of new means to help fight bacterial resistance to antibiotics has turned the attention to nanomaterials in recent decades. The biomedical applications of these innovative structures were initially pointed out, mainly against cancer and non-infectious diseases. Subsequently, research into applications of nanomaterials as antibacterial agents has emerged and researchers focused on their usage to combat biofilms formed by prokaryotic and eukaryotic microorganisms (Table 1).

According to the US National Nanotechnology Initiative’s definition, nanoparticles are defined as “structures with sizes from 1 to 100 nm (1 nm = 10^−9^ m) in at least one dimension”. Their small size, combined with their unique physical and chemical properties, allows them to easily enter the cells of microorganisms and interact with their intracellular components, blocking normal cellular processes. It is crucial that these new antimicrobial agents do not induce resistance, and this is a criterion that so far most nanomaterials have met [74]. This is feasible due to their design to overcome the cellular systems underlying drug resistance [75], including reduced cellular permeability, modification of target molecules, enzymatic inactivation, and inability to export via efflux pumps [76].

In contact with the bacterial membrane, nanomaterials and nanoparticles may cause the disruption of its integrity and leakage of intracellular components (Figure 1). They can insert themselves between individual membrane phospholipids and interact with cellular components such as DNA, ribosomes, and enzymes and disrupt molecular processes. This would accordingly lead to oxidative stress, electrolyte imbalance, and enzyme inhibition, leading to cell death [75].

An important advantage of most nanomaterials is that they can be successfully used as drug delivery carriers, thus further enhancing their potential for future applications [75]. The encapsulation of various antibacterial substances can significantly improve their delivery to the target cells. Additionally, substances transported by drug delivery nanosystems can be released in a controlled or prolonged manner. This approach could remarkably increase their effectiveness, with effects observed even at very low concentrations.

### 5.1. Inorganic Nanoparticles

Inorganic nanoparticles have been extensively studied as antibiofilm agents due to their broad spectrum of properties. Their antimicrobial effect is mainly based on electrostatic binding to the bacterial cell wall and/or release of metal ions. Interaction with the bacterial surface leads to the destruction of the cell membrane and the generation of oxygen radicals. The cell tries to compensate through electron transport and efflux pumps, but this leads to ion imbalance, impaired cellular respiration, disruption of electron transduction, and ultimately cell death. This effect is characteristic and has been observed in numerous studies for various metal nanoparticles, such as silver (AgNPs), gold (AuNPs), and copper (CuNPs), as well as for metal-oxide nanoparticles, e.g., iron oxide, zinc oxide, magnesium oxide, titanium oxide [74,77,78], manganese oxide, etc. [79]. Their antibacterial activity is mainly determined by the structure of the bacterial cell wall. Various studies have shown that metal oxide nanoparticles are more active against Gram-positive than Gram-negative bacteria (Table 1). The reason is that in Gram-negative bacteria, the cell wall contains higher amounts of lipopolysaccharides, lipoproteins, and phospholipids, which creates a mechanical barrier for penetration inside the cells [80]. 

Metal and metal oxide nanoparticles are the subject of a number of studies with direct medical applications. They find considerable success in the treatment of dental caries. Biofilms formed in caries create a local acidic pH microenvironment, promoting the growth of the bacterial species that cause it, which leads to acid dissolution of tooth enamel. Targeted treatment and elimination of caries-causing species is possible with the use of nanomaterials. For instance, nanoparticles composed of ferumoxytol iron oxide nanoparticles (FerIONPs) affect biofilms containing *Streptococcus mutans*. Their mechanism of action involves the interaction of nanoparticles with pathogen-specific glucan-binding proteins and the generation of free radicals [81].

Another possible application is for the treatment of skin biofilm infections. A common cause of these is *S. aureus*. When applied in vitro, ZnO-NPs inhibited the development of existing biofilms of the mentioned species and induced their destruction. In an in vivo study in a mouse model, accelerated wound healing was observed compared to control untreated mice. When combined with gentamicin, the healing time was reduced by almost half [82].

Hybrid systems based on inorganic nanoparticles and antibiotics have been also found to exhibit antibiofilm properties, upgrading the activities of their building components. Recently, DNase–aptamer complex loaded with the β-lactam antibiotic ampicillin (Amp) and zinc oxide (ZnO) nanoparticles was reported, targeting *Methicillin-Resistant Staphylococcus aureus* (MRSA) biofilms. During treatment, ZnO destroys the biofilm. The released bacteria absorb the complex, which releases the DNAse under acidic conditions, cleaving the mecR1 gene, and the ampicillin effectively eliminates MRSA. In vivo assays performed in rabbits showed effective clearance of bacteria and biofilm in the cornea as well as suppression of the proinflammatory cytokines interleukin 1β (IL-1β) and Tumor Necrosis Factor alpha (TNF-α), and showed no toxicity to corneal epithelial cells [83]. In a similar research study, nisin was conjugated with biogenic silver nanoparticles (PchNPs) obtained from extracellular cell-free extracts of *Phanerochaete chrysosporium*. These nanoconjugates were tested for their ability to combat bacterial infections caused by *S. aureus* and *E. coli*. The results demonstrated the presence of antibacterial activity against both strains. However, the PchNPs exhibited a significantly lower inhibitory concentration against *S. aureus* compared to the free form of nisin. This implies a higher level of activity of the complex. These findings are promising and suggest that these complexes may have potential in future antimicrobial applications [84].

### 5.2. Graphene and Its Oxides

Graphene and its oxides are widely studied nanomaterials with known antibacterial activity. Their physical form can damage the bacterial membrane purely mechanically upon contact with it, while chemically, they can disrupt the electrostatic charge by withdrawing electrons from the membrane and disrupting its structure or activating a process that generates oxygen radicals [74]. Through molecular diagnostics, it has been shown that upon the application of graphene oxide nanosheets, they insert themselves into membranes and extract phospholipids, a process known as destructive extraction [85]. In their study, Matharu et al. investigated the possible antibacterial potential of polymer networks containing graphene oxide against *E. coli* K-12. Their results showed a reduction of up to 85% in the bacterial population, with the generation of oxygen radicals in the cells [86]. 

### 5.3. Polymer-Based Nanomaterials

In the past decade, polymer-based nanomaterials have been found to show promising antibacterial properties [87,88]. Polymer-based antibacterials usually carry positively charged functional groups such as amino or quaternary ammonium. Additionally, the advantages of modern polymer chemistry enable the synthesis of more complicated macromolecules that are directly related to the targeted design of novel nanostructures with the desired properties [89,90,91]. 

Chitosan and its derivatives are probably the most studied polymer nanomaterials with promising antibacterial properties. They have high levels of biodegradability and biocompatibility without causing toxicity. Their structure allows for further modification with other natural compounds, antimicrobial metals, and antibiotics, which leads to the enhancement of their effects [92]. Moreover, they exhibit antibacterial activity against a variety of pathogens, e.g., *S. aureus*, *S. epidermidis*, *Listeria monocytogenes*, *E. coli*, *K. pneumoniae*, *Xanthomonas campestris*, and others [93]. A recent study showed good antibacterial and antibiofilm effects of chitosan nanoparticles when applied against clinical isolates of *S. mutans*. They inhibited biofilm formation at a concentration of 0.75 µg/mL [94]. Chitosan can also be used in the form of hydrogels, which can be combined with various nanocomposites. Another study investigated the effect of such a hydrogel (CS) as well as its combination with zinc oxide/zeolite nanocomposite (ZnONC-CS) against *S. mutans* biofilms. The results of the performed tests showed the best antibiofilm effects and reduction in metabolic activity were achieved when applying ZnONC-CS [95]. In another study, the application of hydrophobic chitosan nanoparticles loaded with carvacrol to *P. aeruginosa* biofilms reduced biofilms by 46–53%. Furthermore, the modified nanoparticles significantly reduced the swarming motility of the species by 40–60% [96]. Nanoparticles based on chitosan and poly(D,L-lactide-co-glycoside) nanoparticles loaded with benzalkonium bromide (BZK) aimed at inhibiting and destroying MRSA biofilms are also promising. In vitro test data showed inhibition of biofilm formation at a concentration of 3.33 μg/mL, as well as biofilm disruption at a concentration of 5 mg/mL compared to the aqueous solution of benzalkonium bromide. These nanoparticles caused a significant disruption to bacterial and human cell walls, which was observed through transmission electron microscopy. An in vivo study showed an up to 80% reduction in bacterial cell counts in wounds using BZK nanoparticles for 7 days [97]. A nicotinamide containing chitosan nanoparticles was applied locally in ex vivo experiments for the treatment of Acne vulgaris. These studies demonstrated a strong adherence of the NPs to the skin and a 68% accumulation of nicotinamide in the stratum corneum, epidermis, and dermis. Clinical trials on patients indicated a 73% reduction in inflammatory acne lesions when compared to untreated areas [98].

Poly [2-(dimethylamino)ethylmethacrylate (PDMAEMA) has also been found to be a prospective candidate for antibacterial purposes. It possesses a tertiary amino group in its side chain and easily could be quaternized to quaternary ammonium compound [99,100]. In both forms, PDMAEMA exhibits good antimicrobial effects on a wide spectrum of Gram-negative and Gram-positive strains [69,99,100]. Its cytotoxic effect, however, is related to its functionality as the tertiary amino form exhibits better cellular tolerance than the quaternized ammonium one. Recently, it was shown that through the preparation of mixed polymeric micelles, the cytotoxicity of PDMAEMA-based antibiofilm nanosystems could be controlled. In their work, Stancheva et al. [69] developed mixed nanomicelles bearing in their polymeric shell PDMAEMA and poly(ethylene oxide) (PEO) chains. Toxicity tests revealed a strong composition-dependent cytotoxic effect of the systems, as increasing the amount of PEO in the micellar corona enhanced human diploid fibroblast cell tolerance. The mixed nanosystems were additionally loaded with ciprofloxacine as either loaded or empty they effectively exfoliated mature biofilms of *E. coli* and *S. aureus*. Moreover, they were able to suppress the metabolic activity of sessile bacteria after 24 h treatment. PDMAEMA polymeric micelles bearing AgNPs were found to be effective in the treatment of *Pseudomonas aeruginosa* biofilms [101]. The authors showed that these hybrid particles cause significant biofilm reduction and alteration and the death of biofilm bacteria.

The usage of polymer micelles modified with small molecules, peptides, antibodies, proteins, aptamers, nucleic acids, carbohydrates, or antibacterial agents is an appropriate approach for the treatment of cells infected with various pathogenic microorganisms [102]. Their advantage is the ability to bind to the cellular receptors, triggering endocytosis, followed by the lysis of intracellular enzymes and the release of the active molecule, which disrupts the cells from the inside out [91]. For instance, pH-responsive amphiphilic poly(ethyleneglycol)-b-poly[(3-phenylprop-2-en-1,1-diyl)bis(oxy)bis(ethane-2,1-diyl)diacrylate] micelles were loaded with cinnamaldehyde and used to treat intracellular bacterial infections caused by *S. aureus* in macrophages. After entering the macrophages, through endocytosis, the micelle’s envelope ruptured, and the released cinnamaldehyde attacked the bacterial membrane and disintegrated it. Additionally, the micelles activated the formation of oxygen radicals, thus inducing macrophage differentiation [103]. In a comparative context, aiming to achieve a synergistic effect, other researchers used a poly(vinyl alcohol) (PVA) polymer matrix loaded with AgNPs. The authors prepared a thin layer of PVA-AgNPs and nanofibers that showed an antibiofilm effect against unicellular green algae *P. kessleri*. It was found that the differences in the structure of the thin layer and fibers, as well as the placement of AgNPs inside the polymer matrix, influence the release rate of Ag ions and, accordingly, the interactions of these nanocomposites with cells. The resulting nanocomposites demonstrated established antibacterial and antibiofilm effects and showed potential applications in the field of medicine as well as in the protection of surfaces exposed to moisture or in water [104].

### 5.4. Vesicular Nanosystems

In addition to the listed above, the vesicular nanomaterials could be very promising for antibacterial purposes. This is due to their large empty internal compartment, able to accommodate different antibacterial substances, but also to the similarity of their structure to that of the cellular membrane. They are usually formed from phospholipids, but also from non-ionic surfactants or polymers. Their vesicular and often bilayered structure allows them to enhance skin drug permeation through its effective solubilization [105]. One such potential treatment approach is the application of hexyl-aminolevulinate (HAL)-loaded ethosomes in photodynamic therapy. Their effectiveness was determined in an in vivo experiment on Sprague-Dawley rats infected with *Propionibacterium acnes*. The best results were observed at 5 mg/mL of hexyl-aminolevulinate (HAL) ethosomes. In comparison with the application of pure ethosomes, pure hexyl-aminolevulinate, or 5-aminolevulinic acid, the combination showed excellent therapeutic effects against *P. acnes* biofilms [106]. Another option is the usage of niosomal carriers. Habib et al. designed gel-formed nanoparticles to encapsulate dapsone, an anti-infective drug. Through in vivo experiments performed in a mouse model infected with *Cutibacterium acnes*, the results showed the gel’s penetration to the dermis layer and increased recovery, as well as a significant reduction in inflammation compared to the application of Aknemycin^®^. Each of the presented nanocarriers is a promising drug for the topical treatment of acne [107]. Poly(ε-caprolactone) nanocapsules loaded with carvacrol and thymol showed antibacterial, antifungal, and antibiofilm activities against *S. aureus*, *E. coli*, and *C. albicans*. The encapsulated forms of these essential oils showed higher efficiency compared to the pure ones and a lack of toxicity toward a human keratinocyte cell line [108]. Capsular polysaccharide (CPS) nanoparticles functionalized by amikacin (termed CPS-AM NPs) were tested for their antibacterial and antibiofilm capabilities against *E. coli* and *P. aeruginosa*. The high number of positive charges on CPS-AM NPs enhances their interaction with bacteria, resulting in remarkable bactericidal effectiveness—99.9% for *E. coli* and 100% for *P. aeruginosa*. This outcome is attributed to the damage inflicted to the bacterial cell wall [109].

### 5.5. Antibacterial and Antibiofilm Nanoroughnesses

Recently, a new class of nanomaterials, known as nanoroughnesses, has emerged for antibacterial purposes [110,111,112]. The main concept behind them is related to the structural peculiarities of their surface topography, which can interfere with bacterial adhesion and hence with biofilm formation. Together with roughness, the material the surfaces are made from is very important. Therefore, the physical and chemical properties of nanoroughnesses become essential for their antibacterial activity. For instance, the antibacterial effect of nanorough surfaces was combined with specific sugar metabolites (fructose), thus having the ability to decrease the growth and biofilm formation of *S. aureus* [113]. An example of such different substrates are slippery liquid-infused porous surfaces (SLIPS), which were able inhibit the biofilm adhesion of *P. aeruginosa* with 99.6% efficacy compared to the control sample (superhydrophobic nanostructured polytetrafluoroethylene) [114]. A very comprehensive study in the field was conducted by Nastulyavichus et al. [115]. They studied the effects of a number of parameters, such as laser fluence, the nature of the metal surface (gold, silver, and copper), film thickness, etc., on the antibacterial activity of the prepared nanoroughnesses. The authors used several Gram-positive and Gram-negative bacterial strains. They obtained a pronounced antibacterial property for silver and copper nanoparticles for all strains and, as demonstrated via the EDX method, they did not penetrate mature biofilms but did affect their surfaces and are therefore appropriate for wound treatment. In contrast, gold nanoparticles were found to have no effect on cell viability. Also, laser-synthesized ultrapure silver–gold alloy nanoparticles manifested antibacterial and metabolic activity but with complete diffusion into the matrix of mature *S. aureus* biofilms [116]. In another study, Saraeva et al. prepared nanostructured metal (Au, Ag) films exhibiting antibacterial effects [117] against the pathogens *S. aureus*, *P. aeruginosa*, and *E. coli* [118]. Nanostructures were formed by femtosecond laser ablation, resulting in an array of micro spots. The samples were treated with a low-voltage locally enhanced electric field, resulting in the formation of pores in the membrane and the subsequent apoptosis of bacterial cells together with some alterations in the main cellular components.

### 5.6. Application of Nanomaterials in the Construction of Medical Devices

All implantable medical devices (e.g., cardiac pacemakers, artificial heart valves, artificial joints, hemodialyzers, needleless connectors, urinary catheters, central venous catheters, endotracheal intubation, contact lenses, breast implants, and orthodontic prostheses) carry a risk of biofilm-related infections, which significantly affect patients’ quality of life and even jeopardize it. The development of medical devices on which bacterial colonization is impossible and the capability of treating biofilm infections is an evolving field of application related to nanomaterials. In the process of biofilm development, the most crucial stage is adhesion to the substrate. This process can be inhibited by the application of antibacterial or antiadhesive agents through surface modification of medical devices [108]. Such agents may include metallic and metal oxide nanoparticles, polymeric nanoparticles, carbon-based nanoparticles (such as graphene and nanodiamonds), surfactants, and others [109].

Various examples illustrate the application of nanoparticles in medicine. It has been found that PLGA nanoparticles loaded with paclitaxel (PTX-NP) are promising for application in preventing stenosis at the site of venous injury, even with prolonged use [119]. Zinc oxide nanoparticles synthesized through the application of a medicinal plant extract from *E. odoratum* exhibit promising antibiofilm effectiveness against both Gram-negative and Gram-positive bacteria in venous catheters [120]. As mentioned earlier, a significant percentage of urinary infections are attributed to the high adhesive capability and biofilm-forming activity of Gram-negative bacteria. In such studies, methods allowing for the enhancement of biocompatibility must also be considered. Valuable advancements are associated with improving the antiadhesive and antibiofilm properties of urinary catheters coated with nanoparticles [121,122,123,124]. Moreover, incorporating metal and metal oxide nanoparticles into polymeric membranes is regarded as a natural approach for creating distinctive wound bandages with antibacterial and antibiofilm properties. To facilitate effective wound healing, it is essential to use an appropriate material with superior efficacy that establishes an ideal environment for epidermal regeneration while offering a protective barrier against both water loss and wound infection [125,126,127,128]. Metals and their oxides can be applied alone or as part of complex structured composites. One such possible complex system is a zinc-loaded palygorskite nanocomposite used as a coating on urinary catheters. Nanocomposites were sprayed uniformly on the surface of the catheter with a spray gun. The results of the conducted research showed antibacterial properties and a significant biofilm inhibitory effect on *E. coli* and *S. aureus* [129]. Silver–ricinoleic acid–polystyrene nanoparticles were used for surface modification of polyurethane catheters and showed good antimicrobial and antibiofilm activities against the same two species [130].

A graphene coating doped with silver nanoparticles showed high antibacterial efficiency against *P. aeruginosa* [75]. Graphene oxide can also be applied as a surface coating for biomedical devices. The reduced form of graphene was deposited on aluminum surfaces by sequential functionalization with amines and subsequent immersion in a graphene solution. This led to a covalent attachment reaction of graphene on the treated surfaces. They were then colonized with *E. coli* and the tested strain showed antibacterial activities. This material has also been tested for cytocompatibility against the 3T6 fibroblast target line [131].

A biofilm infection can also develop as a result of wound contamination during a surgical procedure. Various polymeric materials are used in the development of surgical sutures in which bacterial adhesion cannot occur. James et al. developed a biodegradable hybrid polycaprolactone/AgNPs nanomaterial for coating surgical sutures that showed an antibacterial effect against the tested bacterial model of *E. coli* [132].

An antiadhesive coating based on a natural polymer released by a marine cyanobacterium (CyanoCoating) was developed for the protection of urinary catheters. Its effectiveness was evaluated against pathogens responsible for catheter-associated UTIs, including MRSA, *E. coli*, *P. mirabilis*, *K. pneumoniae*, and *C. albicans*, and the results demonstrated an approximately 68–95% antiadhesive efficiency against all tested species [133].

In the manufacturing process of medical devices, it is possible for a material to be engineered to possess multiple properties. The development of nanomaterial contact lenses to prevent and treat keratitis is one such example. Soft contact lenses can be used as a drug delivery system. Silver nanoparticles modified with zwitterionic poly(carboxybetaine-co-dopamine methacrylamide) copolymer were immobilized on lenses by Ma et al. [134]. The lenses showed effective inhibition of biofilm growth in *E. coli*, *S. aureus*, *P. aeruginosa*, MRSA, and *C. albicans* through synergistic action of the zwitterionic surface and the prolonged release of silver ions. Their potential to treat eye infections and prevent tissue disorders was demonstrated in an in vivo rabbit model.

Nanomaterials could also be used in the preparation of dressings for the treatment of biofilm infections in skin wounds. A recently reported system is an injectable sodium alginate hydrogel loaded with plant polyphenol-functionalized silver nanoparticles (GA@AgNPs-SA). Antibacterial and antibiofilm tests demonstrated growth inhibition of *E. coli* ATCC25922, *S. aureus* ATCC6538, and MRSA ATCC29213 on the hydrogel surface. In vivo analysis showed that the GA@AgNPs-SA hydrogel could effectively reduce the expression of IL-6 and TNF-α to alleviate the inflammatory response and promote angiogenesis by regulating the expression of CD31, α-SMA, and VEGF and significantly accelerating the healing process [135]. Another similar system is silver nanoparticles incorporated into genipin-crosslinked gelatin hydrogels used as a wound dressing. The resulting hydrogel demonstrated antibacterial and antibiofilm abilities against *S. aureus*, *Bacillus subtilis*, *P. aeruginosa*, and *E. coli*, inhibiting growth at a minimum inhibitory concentration (MIC) of 63 μg/mL [136].

**Table 1 pharmaceutics-16-00162-t001:** Summary of nanomaterials used for treatment and prevention of bacterial biofilms as well as for construction of medical devices together with their mechanisms and effects on various bacterial strains.

	Nanomaterial	Mechanisms and Effects	Bacteria	Ref.
Polymer-based nanomaterials	Clarithromycin-loaded lipid polymer nanoparticles	Destruction of biofilm extracellular polymeric substances (EPS); antibacterial effects of NPS on biofilm bacteria; inhibition of bacteria adhesion and biofilm formation.	*Helicobacter pylori* SS1	[137]
	Biguanide-derived polymeric nanoparticles (PMET)	Biofilm dispersion; cell surface deformation.	MRSA	[138]
	Chitosan-coated iron oxide nanoparticles	Binding with bacterial membrane through electrostatic interactions and disturbing bacterial cells.	*E. coli* ATCC 35150 *S. epidermidis* ATCC 14990 *Bacillus cereus* ATCC 14579	[139]
	Berberine-loaded chitosan nanoparticles (BBR-CSNPS)	Concentration-dependent inhibition of biofilm formation; destroying cell wall and cell membrane integrity.	*C. albicans*	[12]
Inorganic and vesicular nanomaterials	Chitosan nanoparticles loaded with plant essential oils	Complete exopolysaccharide synthesis arrest; irregular cell shape, smooth cell surface with collapse of nucleolus; loss of cell virulence.	*Acinetobacter baumannii*	[140]
	Zinc oxide (zno) and zinc sulfide (zns) nanoparticles	Killing planktonic bacterial cells; inhibition of biofilm formation.	*S. aureus* ATCC 25923 *Klebsiella oxytoca* ATCC 13182 *P. aeruginosa* ATCC 27853	[141]
	Niosome-loaded selenium nanoparticles	inhibition of biofilm formation.	*S. aureus* ATCC 25923 *E. faecalis* ATCC 29212 *P. aeruginosa* ATCC 39615 *E. coli* ATCC 25922	[142]
	Cinnamon oil-loaded nanoliposomes	Inhibition of biofilm formation; cell surface absorption, penetration, and cell destruction.	*S. aureus* *P. aeruginosa*	[143]
	Hausmannite nanoparticles	Inhibition of biofilm formation.	*P. aeruginosa*	[79]
Nanomatirials for medical devices	Gellan gum-incorporating titanium dioxide nanoparticles biofilm	Inhibition of biofilm formation.	*S. aureus* *E. coli*	[144]
	Acrylic resin-containing silver nanoparticles	Inhibition of biofilm formation.	*C. albicans* ATCC 10231	[145]
	Curcumin–graphene oxide (GO/Cu) surface coating	Blocking cell adhesion; induction of reactive oxygen species (ROS) production.	*Candida parapsilosis*	[146]
	Graphene oxide (GO)/polydimethylsiloxane (PDMS) composites	Increased membrane permeability, metabolic activity, and endogenous ROS synthesis.	*S. aureus* SH1000	[147]

## 6. Natural Compounds for the Treatment of Biofilm Infections

Plants, marine organisms, and microorganisms serve as interesting sources of metabolites with the capability to inhibit or destroy biofilms and suppress quorum sensing (QS) systems. Their effectiveness arises from their diverse chemical compositions. In recent years, there has been a growing recognition of plant metabolites as potential inhibitors of bacterial pathogenesis [148,149,150] (Table 2). The therapeutic properties of plants are primarily attributed to their secondary metabolites, which vary in type and quantity among different plants and exhibit diverse biological activities [151,152]. Natural products have always been a valuable source for developing medications. In fact, more than 80% of the pharmaceuticals available on the market either originate from natural sources or are inspired by natural compounds [153,154,155]. However, the U.S. Food and Drug Administration (FDA) has approved only a few phytochemicals, such as paclitaxel, capsaicin, codeine, colchicine, and reserpine, against drug-resistant bacteria [156].

### 6.1. Purified Phytochemicals

Plants possess substantial quantities of non-nutrient secondary metabolites with bioactive properties, commonly referred to as phytochemicals [157]. They are preferred as antibiofilm and antimicrobial agents because of their relatively non-toxic nature, biocompatibility, and availability [158,159]. Furthermore, phytochemicals can be effective against multidrug-resistant bacteria, including *S. aureus*, *E. coli*, and *K. pneumoniae*, in both their planktonic and biofilm forms [157]. Many phytochemicals act as antagonists of signaling molecules, engaging in competitive inhibition with receptors and inhibiting the QS cascade. Interfering with QS signaling can result in the suppression of biofilm formation by inhibiting the secretion of adhesins and the synthesis of EPSs [160]. Moreover, phytochemicals have the ability to reduce bacterial virulence factors, which are important for microbial invasion, host tissue damage, and evasion of host immunity [161].

Active substances found in plants can be categorized into two primary groups. The first group comprises products of primary metabolism, including carbohydrates (such as sugars and mucous substances), fats (fatty acids and phytosterols), proteins, amino acids, vitamins, enzymes, and pigments. The second group consists of products of secondary metabolism, and their specific activities can vary based on the type of plant or the climate in which they grow [162]. Nowadays, six major groups of phytochemicals are recognized for their significant antibiofilm properties: phenols, alkaloids, terpenoids, polyacetylenes, lectins, and polypeptides. Within these categories, several subgroups can be identified, including phenolic acids, terpenes, essential oils, flavones, flavonols, flavonoids, quinones, tannins, alkaloids, coumarins, isothiocyanates, sulfides, thiosulfinates, and polyamines [160]. Phytochemicals with established antibacterial activity belong to the following chemical classes illustrated in Figure 2: phenols, terpenoids, essential oils, alkaloids, lectins, polypeptides, polyacetylenes, flavonoids, glycosides, steroids, and saponins [163,164].

Phytochemicals inhibit biofilm formation through various mechanisms, as shown in Figure 1. When phytochemicals penetrate the biofilm, they initiate their antibiofilm effects by degrading the matrix structure, forming micropores or microchannels (Figure 3A). They can interact with cell wall proteins (Figure 3B) and disintegrate the phospholipid bilayer of the cell membrane (Figure 3C) [163]. This leads to the disruption of adhesion force (potential) and membrane functions through increasing membrane permeability, causing the accumulation of these compounds in the cytoplasm (Figure 3D) [165]. Consequently, cell lysis occurs, leading to the leakage of intracellular components and subsequent cell death (Figure 3E). Phytochemicals can damage DNA and RNA structures and inhibit various enzymes involved in the replication, transcription, and translation processes (Figure 3F–J). As a result, gene expression, intracellular metabolism, and cell proliferation are inhibited. Their impact on translation leads to impaired synthesis of EPS, QS molecules, virulence factors, and motility structures. They can negatively influence QS systems (Figure 3K) and ATPase activity (Figure 3L) [160,166,167,168]. They can influence efflux pumps (Figure 3M), thereby disrupting proton gradients [160,169]. Together with the formed micropores, this facilitates the entry of various antibacterial substances (Figure 3N). When interacting with the structural proteins of the organelles for motility (Figure 3O) [170], phytochemicals partially suppress bacterial motility, dispersion, and intercellular aggregation [152,164,171]. 

Coumarin and its derivatives are important phytochemical groups with various biological properties. One such derivative is furocoumarin, which has been found to be beneficial in many therapeutic fields, including the treatment of skin diseases, inflammation, and more importantly, CF. In the context of CF, furocoumarins and coumarins have effects on NF-kb, CFTR, and serine proteases [172]. After the COVID-19 pandemic, cases of acute lung injury (ALI) and mortality rates increased. Kolpen et al. found out that in ALI cases, biofilms are the dominant form of bacterial life [173].

### 6.2. Plant Extracts

Plant extracts are complex multicomponent mixtures [174] and are considered one of the largest sources of diverse biomolecules, with different effects on microorganisms, plants, and animals. The extraction of these biomolecules is achieved through applying different methods and solvents [175]. Plant extracts have demonstrated a wide range of activities, including anticancer, antimicrobial, antioxidant [176], anti-inflammatory, and more [177]. Their ability to inhibit bacterial growth and influence bacterial virulence is attributed to the synergistic interactions among the individual components of the extract [178]. This synergy arises from the fact that the different compounds within the extract employ distinct mechanisms of action. Some can suppress resistance mechanisms, while others exhibit pharmacokinetic and physiochemical effects that enhance solubility and availability, lower toxicity, and improve absorption [177].

For years, many plants, such as *Tribulus terrestris*, cranberry, cucumber seeds, etc., have been used to treat UTIs. Cranberry juice is most commonly used in these cases because of sialic acid, which reduces inflammation [179,180]. In another study, garlic was evaluated for its potential preventive effect in an in vivo mice UTI model. The results showed that garlic has the potential to inhibit *P. aeruginosa* and protect the kidneys from tissue damage [181]. Other activities of plant extracts play a crucial role in treating infected wounds. Recently, several in vitro experiments have demonstrated the successful application of plant extracts and EOs during wound healing processes and the treatment of skin diseases. Species from the families *Apiaceae*, *Asphodelaceae*, *Boraginaceae*, *Capparaceae*, *Ebenaceae*, *Fabaceae*, *Lamiaceae*, *Pandaceae*, *Solanaceae*, and others have shown great potential in treating bovine mastitis, skin diseases, and wound healing [182]. Acne vulgaris is a prevalent skin condition. Recent reports have shown that extracts from *Rhodomyrtus tomentosa* (ellagitannins, stilbenes, flavonols, phenolic acids) decrease inflammatory and retentional lesions and inhibit the growth of *C. acnes* [183]. Oral biofilms contain hundreds of different bacteria that can lead to serious diseases within the oral cavity due to biofilm formation. A plant extract from *Iris pallida* was tested on mono- and multispecies bacterial biofilms (*S. aureus* and *P. aeruginosa*) and dental plaque (*Streptococcus gordonii*, *Actinomyces naeslundii*, *Veillonella parvula*, *Fusobacterium nucleatum* subsp. *nucleatum*). The extract proved to be effective against all strains, both in biofilm formation and disruption [184].

### 6.3. Essential Oils

Essential oils are considered important natural products derived from aromatic plants and have been used in traditional medicine for centuries. They can be defined as concentrated hydrophobic liquids containing volatile aromatic oily substances (secondary metabolites) [163,185,186,187,188]. They are biodegradable, affordable, and less toxic than synthetic antimicrobial agents. They are widely utilized in the food and pharmaceutical industries due to their bioactive compounds with antimicrobial properties [187,189]. These oils can be obtained from various plant parts, such as flowers, leaves, seeds, twigs, stems, fruits, roots, and bark [190]. EOs primarily consist of two groups of compounds: terpenoids (monoterpenes, sesquiterpenes, and di-terpenes) and phenylpropanoids [185]. EOs can easily penetrate the cell membrane due to their lipophilic properties, causing a change in its permeability and a reduction in membrane potential, leading to loss of integrity. Inside the cell, they can affect various cell functions, such as membrane transport, synthesis of macromolecules, nutrient processing, and ATP synthesis. EO components can influence enzymatic activity and protein synthesis, inhibiting bacterial metabolism. Moreover, they can cause different cellular components to coagulate. Furthermore, due to their high activity, they can impact the synthesis of virulence factors and their release [191,192]. In addition, they have a high affinity for the biofilm matrix, leading to an increase in their concentration within the biofilm. Essential oils can suppress microorganisms without inducing antimicrobial resistance. Some of their active components act as substrates for the efflux pumps that are usually found in resistant bacteria [163,187,190]. Additionally, they can cause disturbances in proton pump function [188]. Certain EOs can inhibit bacterial cell adhesion during the initial stages of biofilm formation, while others have the capability to suppress QS signaling [189].

During the last few years, different authors have outlined the antimicrobial activities of EOs from various plant sources. Their activity was exhibited against antibiotic-resistant strains, including multidrug-resistant (MDR) phenotypes of *S. aureus*, *Salmonella typhi*, *P. aeruginosa*, *K. pneumonae*, and *E. coli*. In addition to antimicrobial activity, EOs reveal antibiofilm activity against MRSA strains [193]. A substantial portion of the antimicrobial properties of EOs is due to the presence of carvacrol and thymol in them [189]. Many EO components are considered safe by the FDA, which allows for their application in the health, pharmaceutical, cosmetic, and food industries [194].

Several multidrug-resistant (MDR) bacterial species isolated from dogs and cats with UTI infections, such as *E. coli*, *Enterecoccus* spp., and *C. albicans*, were treated with EOs from basil, oregano, anise, and thyme. Thyme and oregano EOs showed high inhibitory activity against all the tested strains. Moreover, another seventy-nine EOs were tested for antibiofilm activity against *E. coli* (UPEC). After isolating the active components of the EOs, it was found that carvacrol and thymol, the main compounds of oregano and thyme oils, significantly inhibited biofilm formation in the UPEC strain. This result was obtained because thymol and carvacrol influenced the fimbrae production and hemagglutinating ability of UPEC [181]. Additionally, EOs from *Myrtus communis* are used in traditional medicine due to their wound-healing properties. These EOs have antioxidant, anti-inflammatory, and antibacterial effects against *S. aureus*, *P. aeruginosa*, *E. coli*, etc. [195]. EOs containing thymol and tyrosol have been proven to be the most effective additives to antimicrobial dressings. In the last few years, encapsulated EOs have gained popularity in wound treatment [183].

**Table 2 pharmaceutics-16-00162-t002:** Phytochemicals for treating bacterial biofilms.

Phytochemical	Mechanism	Effect	Source	Strain	Ref.
Phenols and polyphenol compounds					
Phenolic acids	Bind to proteins on the surface of and inside the bacterial cell; modulate protein flexibility and 3D structure.	Inhibit bacterial adhesion; antibiofilm activity.	Fruits Beverages Whole grains Nuts Spices Seeds Seasonings Tea Cinnamon	*E. coli* *S. aureus* *S. aureus* MRSA *A. baumannii*	[169,178,196]
Flavonoids	Form complexes with soluble proteins, bacterial cell walls, and extracellular components; inhibit the expression of fimbriae; inhibit the activity of helicases during DNA replication; quorum quenching; inhibit efflux pumps; inhibit respiratory chains and ATP production; interact with phospholipids in the cell membrane; negatively influence the synthesis of peptidoglycan and fatty acid synthesis.	Antibiofilm activity; wound-healing effect; treatment of local infections.		*E. coli* O157:H7 *Campylobacter jejuni* *Streptococcus pyogenes* *Yersinia enterocolitica* *Chromobacterium violaceum* *S. epidermidis* *E. faecalis* *Shigella flexneri* *Salmonella* spp. *K. pneumoniae*	[191,193,197]
Tannins	Form complexes with proteins, thus causing the denaturation of enzymes, adhesins, and transport proteins; precipitate metals and proteins; inhibit bacterial cell wall synthesis, disrupt the cell membrane, and inhibit fatty acid biosynthetic pathways; inhibit QS.	Antimicrobial activity; inhibit growth; inhibit multiplication and eradicate bacteria; suppress the expression of virulence factors inhibit biofilm formation; suppress violacein synthesis; inhibit motility.	Walnuts Cashews Nuts Hazelnuts Wine Coffee Tea Grapes Strawberries Blackberries	*S. aureus* MRSA *S. aureus* *P. aeruginosa* *P. mirabilis* *S. epidermidis* *C. violaceum* *Salmonella enterica serovar Typhimurium*	[155,191]
Quinones	Target cell wall polypeptides, adhesin molecules on the cell surface, and membrane-bound proteins, causing protein denaturation.				[155]
Coumarins	Penetrate biofilms, causing EPS destruction and leakage; inhibit EPS synthesis and reduce bacterial motility.	Antibiofilm activity.		*S. aureus*	[191,198]
Alkaloids					
Reserpine		Antibiofilm activity	*Rauwolfia*	*K. pneumonlae*	[199]
Caffeine	Disintegrates bacterial cell membranes; binds with and damages DNA; affects the SOS response.	Strong antimicrobial effect.	*Solanaceae* *Apocynaceae* *Leguminosae* *Rubiaceae* *Fumariaceae*	Gram-positive and Gram-negative bacteria	[197,200,201]
Berberine	Causes DNA intercalation; inhibits RNA polymerase; inhibits DNA gyrase and topoisomerase IV; inhibits cell division.			*E. coli* *P. aeruginosa* *A. baumannii* *S. pyogenes* *S. epidermidis*	[159,193]
Quinolones	Inhibit type II topoisomerase, leading to the inhibition of DNA replication.			*E. coli* *B. subtilis*	[156]
Terpenoids (terpenes)					
Mono-, di-, tri-, tetra-, and sesquiterpenes	Assemble in the lipophilic layer of the cell membrane, changing its fluidity and possibly causing leakage; alter cell morphology.	Cytotoxic and antimicrobial activities; inhibit biofilms; reduce EPS and alginate production; inhibit co-aggregation; delocalize electron system.	Spices (sage, caraway, rosemary, clove, cumin, thyme) *R. officinalis*	*P. aeruginosa* *S. aureus* *L. monocytogenes* *K. pneumonia* *Serratia marcescens* *A. baumannii* *H. pylori* *E. faecalis* *S. aureus* MRSA	[161,192,196,202,203,204]
Carvacrol	Disintegrates the cell membrane; changes fatty acid composition.			*E. coli* *S. aureus* *E. faecalis*	[205]
Saponins	Intercalate into the cell membrane and form complexes with cholesterol; can interact with sugar chains in the membrane, disturbing fluidity and forming pores.				[178]

Plants have become a promising alternative in the battle against bacterial resistance, providing a wealth of bioactive compounds with antimicrobial and potential antiviral properties. Plant metabolites, whether in the form of extracts, essential oils, or pure substances, have displayed significant potential in fighting bacterial infections, especially by targeting the two critical mechanisms of bacterial resistance—biofilm formation and the quorum-sensing system. The knowledge and usage of medicinal plants since ancient times has resulted in additional investigations, leading to the discovery of new plant drug candidates and, in the end, new pharmaceuticals, such as digoxin, paclitaxel, etc. [19,156]. However, most of the available information is based on in vivo tests, which are insufficient and limited. Together with the exponential growth of bacterial resistance mechanisms, the need to discover new antibacterial substances is growing. Hence, it is important to research the distribution, metabolism, biocompatibility, and activity of phytochemicals [201,206].

## 7. Conclusions

The emergence of bacterial biofilms and the resulting persistent and chronic infections have been an attractive area of research in recent years. Simultaneously, there have been rapid developments in nanotechnology and natural medicine, focusing on developing various structures with potential and alternative antibiofilm actions due to their specific mechanisms of targeted activity and resistance prevention. Their applicability, independently or in combination therapy with antibiotics, phages, or others, represents a promising synergistic approach to reducing or preventing the development of infections caused by biofilm formation. The current literature review provides up-to-date information on current developments related to the antibiofilm activities of nanomaterials and plant extracts, aiming to expand the knowledge of and research efforts in preventing and treating biofilm-associated infections and their potential applicability in clinical practice. 

## Figures and Tables

**Figure 1 pharmaceutics-16-00162-f001:**
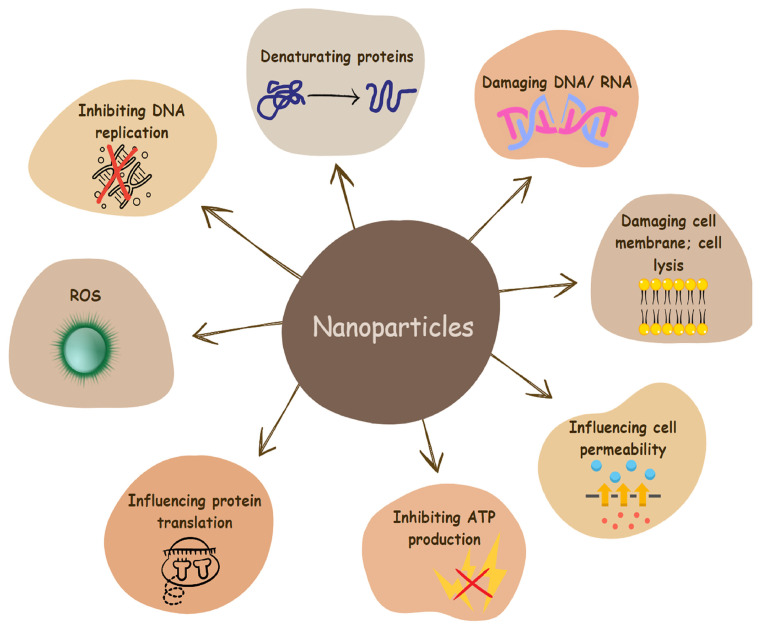
Different mechanisms underlying the modes of actions of nanoparticles on bacterial cells.

**Figure 2 pharmaceutics-16-00162-f002:**
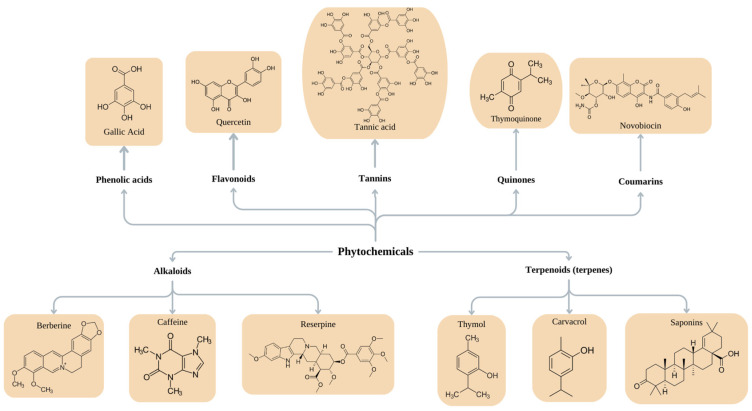
Basic phytochemical structures.

**Figure 3 pharmaceutics-16-00162-f003:**
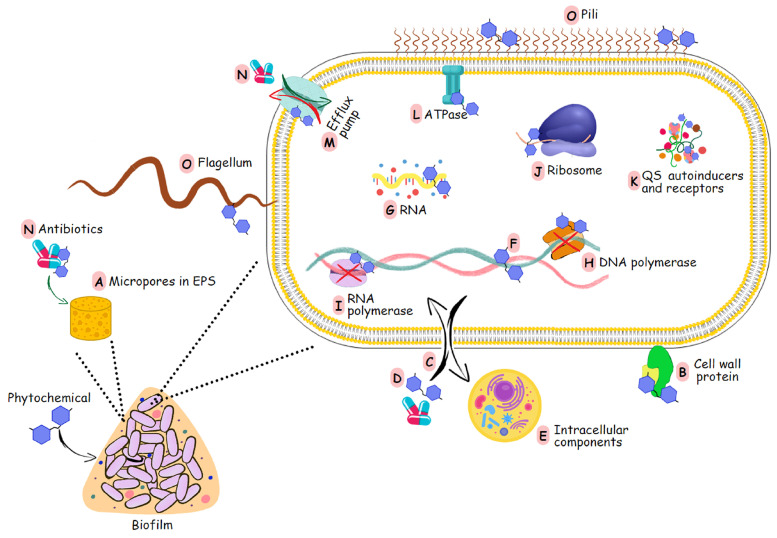
Modes of action by which phytochemicals influence the process of biofilm formation. The phytochemicals act in 15 different ways affecting biofilm and cell structure, as well as cell metabolism and processes as shown (A–O) and as described above. (obtained with copyright permission from Taylor & Francis [160])

## Data Availability

Not applicable.

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
