# Peer review of "An Overview of Biofilm-Associated Infections and the Role of Phytochemicals and Nanomaterials in Their Control and Prevention"

_pharmaceutics, 2024, doi:10.3390/pharmaceutics16020162_

Round 1

Reviewer 1 Report

Comments and Suggestions for Authors

Despite the present rich citations, very considerable sector of antibacterial nanomaterials is missing in the current version and proposed to be included and overviewed. 

Specifically, artificial antibacterial nanoroughness discovered as a strong effect, preventing for a while biofilm formation, by Elena Ivanova (see, e.g., Hasan, J., Crawford, R. J., & Ivanova, E. P. (2013). Antibacterial surfaces: the quest for a new generation of biomaterials. Trends in biotechnology31(5), 295-304 and many others, also from her followers) should be mentioned and discussed.

Next, electrically-activated surface nanoroughness, initiating electroporation effect (see, e.g., Saraeva et al. "Locally Enhanced Electric Field Treatment of E. coli: TEM, FT-IR and Raman Spectrometry Study." Chemosensors 11.7 (2023): 361 and related ones),  should be added for completeness. 

Moreover, laser-induced forward transfer of antibacterial metal films in the form of nanopartciles at high dose onto biofilms (see, e.g., Nastulyavichus et al. "Additive Nanosecond Laser-Induced Forward Transfer of High Antibacterial Metal Nanoparticle Dose onto Foodborne Bacterial Biofilms." Micromachines 13.12 (2022): 2170.) should included and compared to low-dose biocidic nanoparticle procedures (see, e.g., Heine et al. "Anti-Biofilm Properties of Laser-Synthesized, Ultrapure Silver-Gold-Alloy Nanoparticles against Staphylococcus aureus." (2023).). 

Author Response

Thanks to the reviewer for the valuable suggestion. A new subsection 5.5. “Antibacterial and antibiofilm nanoroughnesses” was added (see lines 471 to 499). The proposed articles were cited and included in the references list (ref. No 110 and 115-118).

Reviewer 2 Report

Comments and Suggestions for Authors

The article is an interesting overview of the nature and occurrence of biofilms and the possibilities of combating them. However, the title of the article does not match the content. "Brief" - 32 pages cannot be a "brief". "Innovative approach" - no innovative approach is described in the article, the article provides an overview of where nanoparticles and phytochemicals are already used. I suggest the authors think of a more appropriate title that would more accurately describe the article's content.

I also suggest that they add the following publications to the already cited literature:

1. Preparation, Structure, and Properties of PVA-AgNPs Nanocomposites. Doi: 10.3390/polym15020379

2. Deciphering the photocatalytic degradation of polyaromatic hydrocarbons (PAHs) using hausmannite (Mn­3O4) nanoparticles and their efficacy against bacterial biofilm. https://doi.org/10.1016/j.chemosphere.2023.140961

Why is chapter 3. Biofilm infections associated with the oral cavity separate? Rather, this information would be suitable as subsection 2.4. in chapter 2.

I appreciate the amount of literature the authors have processed into this article, but:

1. Could the authors comment on the effect of the preparation of nanoparticles (biological vs. chemical) on their toxic effect? Is there a difference in toxicity depending on the production method?

2. It would be good if the authors mentioned (at least in some cases) in what form the nanoparticles are used in medicine. They are incorporated into gels, and polymers such as catheters, or are they used as a spray?

3. Could the authors also explain and provide a diagram of the effect of nanoparticles on the cell as shown in Figure 1 for phytochemicals?

Author Response

We have carefully considered all critical remarks, comments and suggestions of the Reviewers and have made all the necessary corrections/amendments accordingly, which are mentioned below. The corrections are outlined with track changes. Our answers and explanations are as follows:

Reviewer 2

  1. The article is an interesting overview of the nature and occurrence of biofilms and the possibilities of combating them. However, the title of the article does not match the content. "Brief" - 32 pages cannot be a "brief". "Innovative approach" - no innovative approach is described in the article, the article provides an overview of where nanoparticles and phytochemicals are already used. I suggest the authors think of a more appropriate title that would more accurately describe the article's content.

Response: Thanks to the Reviewer for this comment. In accordance with the Reviewer's suggestion, we have changed the title as follows: “Biofilm-associated infections: an overview and the role of phytochemicals and nanomaterials for their control and prevention”.  

  1. I also suggest that they add the following publications to the already cited literature:

Preparation, Structure, and Properties of PVA-AgNPs Nanocomposites. Doi: 10.3390/polym15020379 Velgosova.

Response: Thanks to the Reviewer for the valuable suggestion. Additional information was included in the section 5.3. Polymer based nanomaterials as a new paragraph from line 435 to line 444.

  1. Deciphering the photocatalytic degradation of polyaromatic hydrocarbons (PAHs) using hausmannite (Mn¬3O4) nanoparticles and their efficacy against bacterial biofilm. https://doi.org/10.1016/j.chemosphere.2023.140961

Response: The suggested article was added in the section 5.1. Inorganic nanoparticles and also in Table 1 as a new reference [79].

  1. Why is chapter 3. Biofilm infections associated with the oral cavity separate? Rather, this information would be suitable as subsection 2.4. in chapter 2.

Response: Following the Reviewer`s suggestion we have included 2.4. Biofilm infections associated with the oral cavity, as a subsection in the same chapter.

  1. I appreciate the amount of literature the authors have processed into this article, but: Could the authors comment on the effect of the preparation of nanoparticles (biological vs. chemical) on their toxic effect? Is there a difference in toxicity depending on the production method?

Response: In the literature reviewed, not enough information was found regarding the direct relationship between the preparation of nanoparticles and their toxic effect. From our point of view, however, undoubtedly there is such a relationship. The method of preparation of nanoparticles affects their physicochemical properties such as size, shape, morphology, surface potential, and many others. These properties are directly related to the biological behavior of nanoparticles such as interactions with cells, cellular internalization, toxicity etc. As an example we could offer the following: In the case of cationic polymer nanoparticles - the smaller the particles, the higher their total surface area and hence their total surface potential is the stronger. The stronger positive surface potential produces higher toxicity. 

  1. It would be good if the authors mentioned (at least in some cases) in what form the nanoparticles are used in medicine. They are incorporated into gels, and polymers such as catheters, or are they used as a spray?

Response: Thank you for the recommendation to include new information about nanoparticles with biomedical applications. Appropriate information related to the hydrogels was discussed in section 5.6. Application of nanomaterials in the construction of medical devices, but nevertheless the section was extended according to the reviewer`s suggestion by including some additional forms of applications of nanomaterials as new information inserted in the main text of the same subsection.

  1. Could the authors also explain and provide a diagram of the effect of nanoparticles on the cell as shown in Figure 1 for phytochemicals?

Response: The effect of nanoparticles on the cells is described in section 5. “Nanomaterials against biofilm-related infections”. A new figure (Figure 1) was added in the same section, illustrating the mode of action of the nanoparticles.

Reviewer 3 Report

Comments and Suggestions for Authors

The authors have summarised the different nanomaterials and phytochemicals and their role in inhibiting biofilm formation. The topic is of great interest as many patients did not respond to actual treatments, and the need to develop new antimicrobial agents is highly important. The review updates the data published in the last decades, and suitable references are cited.
Pharmaceutics is a flagship journal of MDPI. So, I recommend including a high-quality figure illustrating the action of some of these materials against microorganisms and a scheme with structures of some phytochemicals that have antibacterial activity.
I recommend major revision.

Author Response

We have carefully considered all critical remarks, comments and suggestions of the Reviewers and have made all the necessary corrections/amendments accordingly, which are mentioned below. The corrections are outlined with track changes. Our answers and explanations are as follows:

Reviewer 3

 The authors have summarised the different nanomaterials and phytochemicals and their role in inhibiting biofilm formation. Pharmaceutics is a flagship journal of MDPI. So, I recommend including a high-quality figure illustrating the action of some of these materials against microorganisms, and a scheme with structures of some phytochemicals that have antibacterial activity.

Response: We are grateful to the reviewer for suggesting new figures that would improve the manuscript. In accordance with this suggestion, we performed a new figure (Figure 1) in the section “Nanomaterials against biofilm-related infections“, illustrating mode of action of the nanoparticles on bacterial cells. Regarding the second recommendation we designed a new figure (Figure 2) with structures of some phytochemicals situated in the subsection “6.1. Purified phytochemicals”.

Substantial changes have been made to the text, in line with the recommendation for a major revision

Round 2

Reviewer 1 Report

Comments and Suggestions for Authors

Accept in present form after the revisions made

Reviewer 3 Report

Comments and Suggestions for Authors

The revision of the manuscript has solved all my questions. The authors included the figures suggested to improve the manuscript and adopted all my suggestions. I am satisfied with it. So, I suggest accepting.